# Vaccination Coverage and Predictors of Vaccination among Children Aged 12–23 Months in the Pastoralist Communities of Ethiopia: A Mixed Methods Design

**DOI:** 10.3390/ijerph21081112

**Published:** 2024-08-22

**Authors:** Muluken Dessalegn Muluneh, Sintayehu Abebe, Mihret Ayele, Nuhamin Mesfin, Mohammed Abrar, Virginia Stulz, Makida Berhan

**Affiliations:** 1Amref Health Africa in Ethiopia, Bole Sub City, Woreda 03, P.O. Box 20855, Addis Ababa 1000, Ethiopia; sintayehumoh@gmail.com (S.A.); mihret.ayele@amref.org (M.A.); nuhamin.mesfin@amref.org (N.M.); mohammed.abrar@amref.org (M.A.); makida.berhan@amref.org (M.B.); 2Melbourne School of Population & Global Health, Melbourne University, Carlton, VIC 3053, Australia; 3Faculty of Health, University of Canberra, Bruce, ACT 2617, Australia; virginia.stulz@canberra.edu.au

**Keywords:** vaccination, pastoralist, Afar, coverage, regression

## Abstract

This study assessed vaccination coverage and its associated factors among children aged 12–23 months in pastoralist Ethiopia. It was conducted in three woredas of the Afar region using a community-based cross-sectional mixed methods design with quantitative and qualitative methods. A total of 413 mothers with children aged 12–23 months participated in the quantitative study via a simple random sampling technique. Logistic regression was used to identify factors associated with vaccination, and thematic analysis techniques were used for qualitative data. The percentage of patients who received full vaccination was 25%. Based on vaccination card observations, the dropout rate from Pentavalent-1 to Pentavalent-3 was found to be 2.9%. Logistic regression analysis revealed significant associations between mothers and caretakers with formal education, those who owned mobile phones, had antenatal care (ANC) visits, and birthed at a health facility with full vaccination. The overall proportion of full immunization is lower than the target set by the World Health Organization (WHO). The findings suggest that programs and policy makers should prioritize improving the access and enrolment of women and caretakers, promoting mobile phone ownership, and encouraging ANC visits and the promotion of health facility deliveries, as these are associated with higher rates of immunization.

## 1. Introduction

Immunization remains one of the most important public health interventions and is a cost-effective strategy for reducing both the morbidity and mortality associated with infectious diseases. Over two million deaths are delayed through immunization each year worldwide [1,2]. Despite this, vaccine-preventable diseases remain the most common cause of childhood mortality, with an estimated three million deaths each year in Africa [3,4]. Moreover, a well-functioning immunization system is one of the key elements of a strong health system and prepares a country for future health challenges. In fact, immunization coverage is used as an indicator to assess health system capacity and primary health care access. Investing human, technical, and financial resources in immunization builds the capacity to deliver primary health care services and ensure that no child’s life is jeopardized by a vaccine-preventable disease [5].

After six years of establishment by the WHO in 1974, Ethiopia launched its Expanded Program on Immunization (EPI) in 1980 to reduce mortality, morbidity, and disability in children and mothers from vaccine-preventable diseases [6]. When the program started, the target groups included children under two years of age until it changed to infants and mothers in 1986 to align with the global immunization targets [7]. Since 2019, the primary targets for immunization services in Ethiopia include infants, children between the ages of one and two years for the measles conjugate vaccine (MCV2), girls between the ages of nine and 14 years for the human papillomavirus (HPV) vaccine, and women in the reproductive age group, which is part of the implementation of immunization as a life-course approach. There is an established immunization program support system in the country starting from the federal level down to the health post [8].

With the gradual increase in the incidence of Ethiopia since 2011, the current administrative coverage, as reported in the WHO and United Nations Children’s Fund (UNICEF) estimates of national immunization coverage revised in 2022, reached 89% for BCG, 98% and 93% for diphtheria–pertussis–tetanus (DPT) and DPT3, respectively, 88% for polio, and 89% and 82% for MCV1 and MCV3, respectively [9,10]. A systematic review and meta-analysis study conducted in Ethiopia from 2003 to 2019 revealed that the pooled full immunization coverage in Ethiopia was 58.92%, and the trend of immunization coverage improved from time to time, but there were great disparities among different regions.

According to the Ethiopian Demographic and Health Survey (EDHS), there has been steady progress in EPI coverage over the years. The percentage of children aged 12–23 months who received all basic vaccinations increased from 20% in 2005 and 24% in 2011 to 39% in 2016 and 44% in 2019. Additionally, the proportion of children who were not vaccinated decreased from 24% in 2005 to 19% in 2011 [11]. Despite substantial progress in Ethiopia’s vaccination efforts, disparities in vaccination coverage persist, particularly in underserved and remote regions such as the Afar region of Ethiopia. According to the annual report from the MOH, pentavalent and measles vaccination coverage was low and below the expected targets in the Afar region [12,13]. The percentage of children aged 12–35 months who did not have a vaccination card seen during the home visit was very high in Afar (85.1%), and the lowest percentage had a vaccination history at health facilities in Afar; the percentage who received at least one vaccination at a health facility was 26.2%. The Afar region is a percentage pastoralist community characterized by unique sociodemographic and geographical features, including a predominantly pastoralist population, limited access to healthcare services, and harsh environmental conditions. These factors can significantly impact healthcare-seeking behaviors, vaccine accessibility, and healthcare delivery strategies in the region [14]. Additionally, access to healthcare services is limited, and infrastructure barriers pose significant challenges to vaccine delivery and uptake.

Religion can significantly influence health behaviors, including decisions related to vaccination. In many communities, religious beliefs and practices shape attitudes toward health interventions, often determining whether individuals and families choose to vaccinate their children. Some religious groups view vaccination positively, seeing it as a means to protect health, which aligns with their beliefs about caring for the body as a sacred responsibility [15]. Conversely, other groups may express hesitancy or resistance toward vaccination based on religious or cultural reasons, which can create barriers to immunization [16].

In Ethiopia, a country with a diverse religious landscape, the impact of religion on vaccination practices is particularly pronounced. Religious leaders often hold significant influence over their communities and can sway public opinion on health matters. In some areas, including the Afar region, religious and cultural factors may contribute to lower vaccination rates. The predominantly Muslim population in Afar may have unique religious practices and beliefs that affect their healthcare-seeking behaviors, including the acceptance or refusal of vaccines [17]. 

Engaging religious leaders in public health campaigns and incorporating religious teachings that support vaccination can help increase vaccine acceptance in communities where religious beliefs are a significant factor. 

Various studies conducted in different countries have shown that obstetric factors, maternal literacy, maternal knowledge, maternal attitude, having at least two ANC follow-ups, postnatal check-ups within two months after birth, place of birth, and short distances from vaccination sites were significantly associated with complete immunization [9,10]. Previous studies have examined vaccination coverage and predictors of immunization uptake in different regions of Ethiopia. However, there is limited existing research conducted in Afar that overlooks the unique sociocultural, economic, and geographical factors that may influence vaccination behavior in this context. Moreover, unlike many previous studies that may have relied solely on quantitative approaches, this study employs a parallel mixed methods design, incorporating both quantitative and qualitative methods. This methodological approach enhances the robustness of the findings by triangulating data from different sources and providing a deeper understanding of the factors influencing full immunization in Afar contexts. Therefore, there is a clear gap in the literature regarding the determinants of vaccination among children in the Afar region. Consequently, this study aims to fill the gap left by previous studies by providing both qualitative and quantitative research. Therefore, this study aimed to assess vaccination coverage, the dropout rate and associated factors among children aged 12–23 months residing in the Afar region, Ethiopia. 

The primary objective of this study is to assess vaccination coverage, the dropout rate, and the factors associated with full immunization among children aged 12–23 months residing in the Afar region, Ethiopia. This research is significant because it addresses a clear gap in the literature, focusing on the unique sociocultural, economic, and geographical factors that influence vaccination behavior in this under-researched area. Some of the key things that stand out about how this study contributes new insights include the focused examination of vaccination in a pastoralist population using mixed methods, the identification of specific predictors, and the geographic concentration on the understudied Afar region of Ethiopia. This contributes important new knowledge to improve vaccination access and uptake in hard-to-reach communities. By employing a mixed methods design that integrates both quantitative and qualitative approaches, our study aims to provide a comprehensive and nuanced understanding of the determinants of vaccination in the Afar region, ultimately contributing valuable insights that can inform targeted interventions to improve immunization coverage in similar contexts.

## 2. Materials and Methods

### 2.1. Study Area and Design and Population

The study was conducted in the Elidar, Dubti, and Gerani woredas of the Afar region. The Afar region is one of the regional state administrations in Ethiopia, with 34 woredas. Semera town is the regional capital of the Afar region, which is 591 km from Addis Ababa. A community-based, cross-sectional study design was used to undertake the quantitative study. A descriptive, qualitative-study design approach was used for the qualitative study. The three districts have a total population size of 232,033. Elidar has 105,070 inhabitants, Dubti (85,575), and Gerani (42,388). All mothers/caretakers who have children 12–23 months of age in selected woredas of the Afar region were selected. All mothers/caretakers who had children who fulfilled the inclusion criteria were selected by a simple, random sampling technique during the data collection period. Mothers/caretakers who had children 12–23 months of age and who were found in their home during the data collection period and who resided for at least one year in the study area were eligible for the study. The study was conducted between December 2023 and January 2024.

### 2.2. Sample Size Determination and Sampling Procedure

For the quantitative part, the sample size of the evaluation was determined using a single population proportion formula considering the objectives of the baseline study of the project. To obtain the maximum sample size, 50% of the immunization among 12–23-month-old children was taken with a 95% level of certainty and a 5% margin of error. Considering a 10% non-response rate, the final sample size was 419. The samples were distributed in proportion to their population size to three districts. For the qualitative part, a total of nine key informant interviews (KIIs) were conducted among woreda health office EPI focal persons, health care workers (HCWs), and health extension workers (HEWs).

This sample was withdrawn from the three identified districts; in each district, three to four kebeles, the smallest administrative units in Ethiopia, were randomly selected using the WHO-recommended survey strategy approach for immunization. Once the kebeles were identified, the Gote/villages were mapped. In the village, all candidate households (12–23 months and mother/father/caretaker pairs) were listed using data from the HEW’s data registry document, and simple random sampling approaches were used to select study participants. The sample was distributed to each district based on the population size (Elidar = 190, Dubti = 155, and Gerani = 74). For the qualitative part, a purposive sampling procedure was used to obtain FGD and KII study participants.

Note that the results from our study in the Afar region are not directly extrapolable to the entire country of Ethiopia due to significant regional differences in sociocultural, economic, and geographic contexts. Ethiopia is a highly diverse country with various regions characterized by distinct lifestyles. Afar, as a predominantly pastoralist and semi-nomadic region, presents unique challenges that may not be present in other parts of Ethiopia, such as the high mobility of its population, limited healthcare facilities, and unique cultural practices. These factors heavily influence vaccination behaviors and coverage in ways that differ from other regions, such as the urban centers or agrarian communities in other parts of the country. The specific findings from Afar may not be fully generalizable to all of Ethiopia; they provide critical insights into the challenges faced by other pastoralist and remote communities within the country. These insights are valuable for understanding and addressing vaccination coverage in similar contexts both within Ethiopia and in other countries with pastoralist populations.

### 2.3. Study Variables and Operational Definitions

In this study, immunization coverage was a dependent variable. Additionally, the independent variables were the sociodemographic characteristics of mothers (age, residence, marital status, education status, occupation, partner’s education status, total family size), maternal health care utilization status, and mother’s knowledge of immunization, and the child-related variables included the number of live births, sex of index child, age of indexed child in months, order of indexed child, index childbirth condition, and index child living with partner. Full immunization coverage: calculated when a child has received the BCG vaccine, three doses of pentavalent, three doses of oral polio, three doses of PCV, two doses of Rota, a dose of IPV, and one dose of measles before the age of one year [18].

### 2.4. Data Collection and Management

For the quantitative part, data were collected using a questionnaire adopted from the WHO, Ethiopian Demographic Health Survey (EDHS), and other pertinent literature and then translated into Amharic. Mothers or caretakers were asked to present vaccination records for an indexed child (a child included in the study). The information about vaccination was then copied into the study instrument (questionnaire). Additionally, BCG scars were also checked on the child’s body. If a mother or caretaker reported that her child did not have a vaccination card, she was asked questions that were intended to be answered about the child’s vaccination. Different strategies, such as the site of vaccination given (oral, injection, and scar), the age at which the child received a specific vaccine, and the ability to distinguish routine vaccination schedules from campaign vaccination, were provided by the data collectors to help mothers/caretakers remember the vaccine taken by the children and to minimize recall bias. The household heads were interviewed based on the structured questionnaire. The data were collected electronically using the mobile applications Open Data Kits (ODK) and Kobo, where a structured questionnaire with pre-coded answers was uploaded.

The qualitative part of the study was addressed using KIIs. KIIs were performed among HCWs, HEWs, and health development armies. A semi-structured interview guide was used to facilitate in-depth interviews with the key informants to obtain insightful qualitative information on key issues to be examined in this baseline study. The interviews were facilitated by two individuals, one as a facilitator and the other serving as a notetaker. All interviews were tape-recorded after consent was obtained from each participant. Key informants were selected purposively based on their experience with the EPI program.

To ensure the data quality of the study, the team used highly consulted, client-approved, and pretested data-collection tools developed through reviewing the relevant literature/national standards. In addition, the recruitment of experienced data collectors and supervisors considered their ability to speak the local language of the study sites. The training was conducted with the aim of developing a common understanding of the study objective and tools.

### 2.5. Data Processing and Analysis

The data were cleaned and transferred to SPSS version 25 for further analysis. The quantitative data from the community survey were summarized using descriptive statistics. A comparison of selected indicators was performed using relevant analysis criteria. Factors associated with vaccinations were assessed using binary logistic regression. Independent variables with a *p* value < 0.20 were candidates for inclusion in the multivariable logistic regression analysis. In the final model, a statistically significant association with the vaccination status of the children was declared using *p* values less than 0.05, and a 95% confidence interval (CI) of the AOR was considered in all cases.

The qualitative data were transcribed verbatim in local languages and translated to English by language experts. Then, the translated material was read to understand the importance of the material. Finally, the transcribed and translated data were analyzed via thematic analysis.

## 3. Results

A total of 419 sampled children aged 12–23 months were included, for a response rate of 100%.

### 3.1. Sociodemographic Characteristics of Respondents

The majority of the sampled children were from the Elidar woreda (45.3%), followed by Dubti (36.8%). More than three-quarters of the participants (328, 78.3%) resided in rural areas. More than two-thirds of mothers (142, 33.9%) were in the age range of 22–25 years, followed by those aged ≥30 years (117, 27.9%). Almost all respondents were married (408, 97.4%), and most of the respondents were unable to read and write (345, 82.3%). Regarding religion, more than three-quarters of respondents (79%) were Muslim and were housewives (331, 79%). Concerning household economic status, most respondents reported their economic status as able to meet basic needs (343, 81.9%), followed by being unable to meet basic needs without charity (52, 12.4%) (Table 1).

### 3.2. Child Characteristics

Less than three-quarters of the mothers had a gravidity of four or fewer (302, 72.1%). Similarly, more than three-quarters (332, 79.2%) of them had a parity of less than or equal to four. More than half (235, 56.1%) of the children were males. The mean age of the index children was 16.7 months, with a standard deviation of 4.05 months. Almost one-third of the children (127, 30.3%) were in the age range of 12–13 months, followed by those in the age range of 21–23 months (103, 24.6%) (Table 2). All the children had mothers as their primary caretaker (419, 100%).

Almost three-quarters (313, 74.7%) of respondents heard about childhood vaccines. Two hundred and eighty-seven (90.7%) participants mentioned that vaccination is important for protecting children from diseases, and participants also mentioned that vaccination is advantageous for healthy children (224, 71.6%). On the other hand, respondents also mentioned the disadvantages of vaccines. More than two-thirds (67%) of them reported that vaccines have side effects and that vaccines may make children sick (56.9%) (Table 3).

### 3.3. Factors Related to Vaccine Refusal and Vaccination Service Satisfaction

Regarding vaccine refusal, 52 (21.8%) respondents had a history of vaccine refusal for different reasons. Among the reasons included being given too many vaccines during the vaccination visit (28, 53.8%), a history of child illness after vaccination (26, 50%), and a fear of injection pain (21, 40.4%). The least mentioned reason for refusal was the mother’s or caretaker’s misunderstanding that a single immunization qualifies as a full vaccination (4, 7.7%). Fear, doubts, and suspicions about vaccines were reported among a small number of respondents (27, 8.6%), while the majority of mothers and caretakers did not have doubts or suspicions about vaccines (278, 88.8%). The reasons for fear, doubts, or suspicions about vaccines were vaccination side effects (15, 55.6%), vaccinations that can make children sick (8, 29.6%), vaccinations that can sterilize children (2, 7.4%), and other factors, such as politics and religious concerns (2, 7.4%). Most of the respondents had good attitudes toward vaccines, as the study participants reported receiving vaccine recommendations to other community members (271, 86.6%).

The study participants reported different reasons for dissatisfaction; of these, the most common reasons mentioned were inaccessible vaccination services (90, 88.2%), vaccination site closed/vaccinator absent (67, 65.7%), no available vaccine at the vaccination sites (55, 53.9%), and long waiting times (58, 56.9%) (see Figure 1).

### 3.4. Maternal Health Service

Over half of the mothers (240, 57.3%) had at least one ANC follow-up, and nearly two-thirds (143, 59.6%) had no more than three ANC visits. Almost two-thirds of ANC visits were in health centers (152, 63.3%). Most of the mothers were informed about child vaccination during their ANC follow-up (Table 4). More than half of the children (227, 54.2%) were born at home. Regarding post-natal check-ups, only one-quarter (103, 24.9%) had check-ups after birth. Most of them were informed about vaccination during post-natal check-ups (97, 94.2%) (Table 5).

### 3.5. Vaccination Coverage

The overall percentage of children aged 12–23 months who had ever been vaccinated was 58.9% (*n* = 247), with 95% CIs ranging from 54.2% to 63.7%. The percentages of children aged 12–23 months who had ever been vaccinated in Elidar were 71.6% (*n* = 136), 46.1% (*n* = 71), and 53.3% (*n* = 40) in Dubti, Dubti, and Gereni, respectively.

#### 3.5.1. Proportion of Vaccination Based on Card Observation

The proportion of patients who received full vaccination via child vaccination cards was relatively greater at 70.7%, with corresponding 95% CIs ranging from 60.4% to 79.1% (*n* = 65). The overall full vaccination status of all sampled children was 15.5%, with a 95% CI ranging from 12% to 19%. The dropout rate from Pentavelnt-1 to Pentavalent-3 was found to be 2.9% (*n* = 12) (Table 6).

#### 3.5.2. Proportion of Vaccination without Card Observation

The percentages of mothers or caretakers reporting vaccines in the absence of a vaccination card were 78.7% (*n* = 122) for BCG, 56.2% (*n* = 86) for three rounds of polio, 57.8% (*n* = 85) for three rounds of pentavalent, 57.2% (*n* = 83) for three rounds of PCV, 92.1%% (*n* = 129) for two rounds of rota, 67.1% (*n* = 104) for IPV, and 92.9% (*n* = 144) for measles (see Figure 2). Based on the vaccination card observations, the dropout rate from Pentavalent-1 to Pentavalent-3 was found to be 2.9%. Note that in the figure below, “cardless” refers to vaccination where there is no card at the time of visit, but we consider the self-reported information from the caregiver or mother stating that the child is vaccinated, even though it has not been verified by the observation of the card.

#### 3.5.3. Proportion of Full Immunization

The proportion of full immunization for basic vaccines among children aged 12–23 months was found to be 25.1%, with a 95% CI ranging from 20.9 to 29.2%. The proportion of full vaccination was not equal across the three districts. The proportion of full vaccination among rural children was 37.4% with a 95% CI of 30.5–44.2% in Elidar, 25.3% with a 95% CI of 15.5–35.2% in Gereni, and 9.7% with a 95% CI of 5.1–14.4% in Dubti. The proportion of measles among 15-month-old children was 111 (44.2%). The proportion of BCG scars according to the data collector’s observations was 34.4% (*n* = 144). The following were significantly associated with full vaccination rates: having formal education 3.90 (1.53–9.98), owning mobile phones 2.99 (1.33–6.76), attending an ANC visit 2.39 (1.14–5.01), and birthing in a health facility 5.79 (2.77–12.12) were positively and significantly associated with full immunization.

### 3.6. Barriers to Child Vaccination

Many different barriers were mentioned by the respondents; of these, the most common were not visiting the village (247, 58.9%), having a domestic workload (180, 43.0%), not accessing a vaccination service (155, 37%), having a closed/vaccinator vacancy at the vaccination site (153, 36.5%), having a long waiting time (138, 32.9%), and not receiving a vaccine at the vaccination site (106, 25.3%).

Regarding vaccine approval status, respondents revealed that most of the people around their community setting approved of vaccinations (308, 73.5%) and that vaccine approval was made by husbands/partners (301, 97.7%), followed by parents/parent in-laws (134, 43.5%) (Table 7).

The qualitative findings support that some of the key barriers to vaccination included the distance of the health facility, the lack of health professional motivation, the desert weather conditions of Afar, and the community and partner engagement, which are some of the reasons for the low coverage of immunization.

Distance: Various KIIs stated that one of the reasons for low vaccination coverage is the long distance to the nearest health facility. The health center EPI focal person described the problem related to long distances to health facilities as follows:

“*The health centre is far away from the majority of kebele residents. Those community members close to this health centre received frequent services from the institution, while those farther away from it did not often access the service. This scenario also applies to other health facilities in the region. Therefore, the long distance to the nearest health facility is the main barrier to EPI services.*”

Similarly, the woreda EPI focal person emphasized long distances to the health center and other potential vaccination service sites, and a lack of transportation made the service inaccessible and unavailable to all parts of the region.

“*The biggest problem for desert or Berhama areas such as the Afar region is the lack of health facilities, such as a nearby health center, which makes vaccination services inaccessible and creates problems in the supply of vaccines.*”

The Dubti Woreda Health Office focal person described the impact of the COVID-19 pandemic on the routine EPI program:

“*When COVID-19 first began to spread in our nation, we had delays in implementing routine EPI delivery services. Later, when the COVID-19 epidemic began to decline, we promptly resumed our normal EPI activities integrated with COVID-19 prevention.*”

Health workers’ motivation: All of the key informants highlighted reasons for low vaccination and the main challenges that prevent children from receiving vaccines.

“*The major barriers to not vaccinating children in our communities include a lack of commitment among health care workers, a shortage of EPI logistics and distribution, a lack of transportation access and high staff turnover.*”

The Gereni Woreda Health Office EPI focal person stated the importance of motivating health workers as a lack of incentives and other motivating things that are not implemented. In addition, these health centers should be constructed because the number of health facilities in these areas is limited. Moreover, conflict is also mentioned as one of the reasons for low vaccination coverage.

“*There is a lack of incentive for health workers, HEWs, and women’s developmental army. The provision of incentives is the best way to motivate people and increase the performance of activities. In our cases, we tried this approach, and we achieved better results. However, it was not enough and was not supported by health higher officials. In addition, our woreda is a geographically vast, populated, and repeatedly conflict-affected area. The woreda had only one health centre, which made it very distant from three health posts. This made it very difficult to conduct the expected follow-up and support on EPI services across different catchment areas. Therefore, additional health centres should be constructed, and conflict issues should also receive attention and should be resolved permanently.*”

Program integration and partner engagement: Study participants also described the importance of integrating stakeholders and the need for governmental commitment to vaccination services. Informants from the woreda health office focal person and health center EPI focal person spoke about this issue as follows:

“*In this woreda vaccine service, delivery strategies are implemented only with the initiative of NGOs, and this alone couldn’t solve our community’s problem at large. Therefore, itis essential to integrate woreda political leaders, community influential leaders and other concerned stakeholder leaders to be involved in vaccine service delivery strategies.*”

The Elidar Woreda Health Office focal person described partners involved in the immunization coverage program as well as areas of improvement in the potential collaboration of stakeholders.

“*Amref Health Africa is an NGO partner that is engaged in promoting the expansion of EPI vaccination coverage. Additionally, Amref and other stakeholders should support us build health posts in five difficult-to-reach kebeles in Woreda. Consequently, this will help us increase immunization coverage.*”

Similarly, HEWs also described NGO partners who were involved in supporting vaccination activities. One extension worker described this as follows:

“*Amref, supported us on implementation of capacity building of voluntary communities Woreda level review meetings and the EPI Vaccination program. The Woreda Health Office supported us in vaccine logistics supply and transportation, but it is not enough if any partner supported us with a transportation vehicle with fuel; it would help us to enhance our vaccination coverage.*”

Community engagement: According to the EPI focal people from different health facilities, they described the importance of community engagement as a new practice for improving vaccination programs.

“*The community representatives should participate during the planning and implementation of immunization activities. For example, in deciding the outreach sites, target identification, and arrangement of the services. Therefore, their participation will help us to achieve better vaccination coverage.*”

### 3.7. Factors Associated with Full Immunization among Children

Mothers and caretakers who had received formal education were almost four times more likely to have vaccinated their children than those who had not received formal education [Adjusted Odds Ratio (AOR) = 3.90; 95% CI: (1.53–9.98)].

The presence of mobile phones at home was found to be significantly associated with the vaccination status of children, as the odds of being vaccinated were three times greater among study participants who owned mobile phones than among study participants who had no mobile phones at home [AOR = 2.99; 95% CI: (1.33–6.76)].

The odds of being vaccinated were 2.39 times greater among mothers who had attended at least one ANC visit than among mothers who had not attended an ANC follow-up [AOR = 2.39; 95% CI: (1.14–5.01)].

Mothers who gave birth at a health facility were 5.79 times more likely to vaccinate their children than mothers who gave birth at home [AOR = 5.79; 95% CI: (2.77–12.12)] (Table 8).

## 4. Discussion

This study assessed vaccination coverage and its associated factors among children aged 12 to 23 months in three woredas in the Afar region of Ethiopia. The full immunization status of the children was assessed using vaccination cards and the mother’s recall method.

The proportion of full vaccination was found to be 25.1% among children aged 12 to 23 months. The immunization coverage across the three woredas differed, which is related to the difference in study participants or respondents, as there was a greater proportion of urban residents in the Elidar and Gereni woredas. This finding is greater than that of the Afar regional state being fully vaccinated according to the Mini-EDHS report of 2019 and another study conducted in the Amibara District of Africa in 2013 [10,11,19]. Similarly, based on systematic and meta-analysis reports of vaccination coverage studies in Ethiopia, this percentage is greater than the proportion of full vaccination in the Afar region, which was 21% (95% CI: 18, 24%)] [20]. This difference could be due to the study setting of the EDHS survey and the improvement of vaccination over time, as there is a time difference.

However, this finding is lower than that of other vaccination coverage reports from other parts of the country. A systematic meta-analysis of vaccination coverage studies in Ethiopia was performed. The overall prevalence of full vaccination coverage among children in Ethiopia was 60% (95% CI: 51, 69%) [20]. Moreover, this finding is less than that of studies conducted in other parts of the country. This could be related to the known weather conditions of the region and the national security issues that could also contribute to this low coverage status. The qualitative findings indicated that distance to health facilities is the main barrier to vaccination services throughout the region. The greatest problem for desert or Berhama areas such as the Afar region is the lack of health facilities, which makes vaccination services inaccessible and creates problems in the supply chain of vaccines. Moreover, pastoralists’ mobile lifestyle is also one possible reason for low vaccination coverage. This indicates the need for integrated mobile and outreach immunization services for hard-to-reach areas, especially pastoral and semi pastoral regions.

The proportion of MCV1 among children aged 12–23 months was 54.4%. The proportion of second-dose measles among children aged 15–23 months was 44.2%. This finding is lower than the WHO-recommended proportion of patients receiving second-dose measles.

Mothers’ and caretakers’ educational status was found to be significantly associated with full vaccination status, as mothers and caretakers who had attended formal education were almost four times more likely to vaccinate their children than those who did not attend formal education. This finding was consistent with other studies performed in other areas of the country [7,11,21]. The findings suggest that complete immunization was more common in children whose mothers were educated, partly because maternal education leads to the acquisition of literacy skills that can be applied to knowledge of vaccination and child protection and better health-seeking behaviors, which then improves immunization uptake for their children. This suggests that educated mothers are better informed about the importance of vaccinations and the negative impact of unvaccinated baby, the diseases they prevent, and the schedules for immunization; consequently, they are more likely to access healthcare services and more proactive in seeking out and adhering to immunization schedules. 

The odds of being vaccinated were three times greater among study participants who owned mobile phones. This could be an important finding, as making contact with mothers for appointments can be facilitated through mobile phones. Multiple studies have indicated that mobile health (mHealth) interventions can improve access to MCH services. A meta-analysis in developing countries indicated that mobile phone reminders were beneficial [18,22]. This might have attributed to our findings where, in Ethiopia, challenges such as geographic barriers, limited healthcare infrastructure, and low literacy levels often hinder vaccination efforts; therefore, mHealth initiatives can play a crucial role. By providing timely SMS reminders to parents about upcoming vaccination appointments, mHealth can help reduce missed opportunities and dropouts, especially in remote areas. These interventions can also deliver culturally relevant educational content, increasing awareness and the understanding of vaccines, which is essential in combating misinformation. Furthermore, mHealth can facilitate real-time data collection and tracking, allowing healthcare providers to monitor vaccination coverage and quickly identify regions with low uptake. By improving access to information and services, mHealth can help overcome the unique barriers faced in Ethiopia, leading to higher vaccination rates and better overall health outcomes.

The odds of being vaccinated were 2.39 times greater among mothers who had attended at least one ANC visit than among mothers who had not attended an ANC follow-up. This finding is in agreement with other studies conducted in Ethiopia [19,22,23]. This could be related to the counseling given at health facilities during ANC follow-ups, which may bring about the desired behavioral change toward vaccines. An ANC follow-up provides an opportunity to promote health care utilization, including institutional delivery, PNC, immunization, and family planning. According to a propensity score matching analysis on strategies to improve child immunization via ANC, antenatal clinics are the conventional platforms for educating pregnant women on the benefits of child immunization [24]. This attributes to the ANC visits, during which healthcare providers often educate expectant mothers about the importance of child immunization, the schedule of vaccines, and the diseases they prevent, linking mothers to the healthcare system and facilitating follow-ups for post-delivery vaccinations. These visits often integrate other maternal and child health services, fostering continued access to healthcare, which increases trust and confidence in the system. Additionally, ANC services promote institutional deliveries, which are associated with higher immunization rates, thereby indirectly improving overall vaccination coverage.

Mothers who gave birth at a health facility were more likely to vaccinate their children than mothers who gave birth at home. This finding is in line with findings from previous studies showing that institutional birth increased the chances of children being fully vaccinated [19,23]. Mothers may be advised and receive education to use postnatal care and vaccination services during institutional birth. Women who give birth in health facilities may also be more inclined to use preventive services, such as infant immunization. The greater likelihood of receiving a complete immunization may also be attributable to the prompt administration of the BCG vaccine following childbirth and vaccination advice received at a medical facility.

The unique challenges faced by pastoralist communities in Ethiopia, such as geographic isolation, nomadic lifestyles, and limited access to healthcare infrastructure, create substantial barriers to routine immunization. These challenges are well-documented in studies of similar populations, where the combination of mobile, remote locations and the scarcity of healthcare facilities complicates efforts to deliver consistent health services [25]. These barriers underscore the necessity of targeted interventions that are specifically designed to address the distinctive needs of these communities.

One critical area for intervention is the educational gap that often exists in these regions. Culturally tailored health education programs that engage local leaders and utilize local languages are essential for effectively communicating the importance of vaccination. Such programs have been shown to improve health outcomes by increasing awareness and changing attitudes toward immunization within similar contexts [26]. By involving respected community figures and delivering messages in a culturally relevant manner, these programs can significantly enhance the uptake of vaccinations.

In addition to educational initiatives, improving access to healthcare services through innovative delivery methods is crucial. Mobile clinics and outreach programs have been successfully implemented in other low-resource settings to bring essential services directly to hard-to-reach populations. Moreover, mHealth initiatives—such as mobile phone reminders and health information dissemination—offer a promising way to maintain contact with nomadic populations and ensure timely vaccination [27]. By leveraging technology and flexible service delivery models, healthcare providers can overcome many of the logistical challenges associated with serving pastoralist communities.

### Policy Implication

Our study provides new insights into how specific regional factors within Ethiopia, particularly in the Afar region’s pastoralist communities, influence vaccination coverage. Unlike the studies cited, which focus on relatively more stable settings with established healthcare systems, our research highlights the challenges faced in a region characterized by nomadic lifestyles and significant geographic and socioeconomic barriers. Moreover, in the pastoral context, such as in the Afar regions, there is a scarcity of evidence globally, particularly in the Ethiopian context. The findings from these regions are highly specific and only applicable in this context. For instance, mHealth (mobile health) is an appropriate approach for these mobile communities where it is challenging to provide services in a static context; this might be different where the context is different. Furthermore, we elaborate on how the unique socio-cultural and logistical conditions of the Afar region contribute to the findings, providing a comparative analysis with other regions and settings. This approach underscores the specific context of our study and its contributions to the broader understanding of vaccination coverage in underserved, nomadic populations. In line with this, we included a detailed discussion of the implications of these regional differences for policy and practice, making the case for tailored interventions that address the unique needs of such communities. This should enhance the paper’s originality and relevance.

The attached study on vaccination coverage and predictors of vaccination among children aged 12–23 months in the pastoralist communities of Ethiopia highlights several policy implications to improve immunization rates and address the unique challenges in the Pastoralist context. Based on this study, this includes enhancing the availability and accessibility of healthcare services, particularly in remote and underserved areas like the Afar region. This study has an implementation for program implementors and decision makers either at policy or implementation level in the following areas: (i) improving the infrastructure of health facilities and increasing the number of healthcare workers; (ii) prioritize education for mothers and caretakers, as it is a significant predictor of vaccination coverage. Programs aimed at increasing maternal education can help improve health literacy and decision-making regarding vaccinations; (iii) implement mHealth programs to facilitate the dissemination of health information, appointment reminders, and follow-up care; (iv) strengthen ANC services to ensure that pregnant women receive comprehensive care, including information about vaccinations; (v) encourage women to give birth at health facilities where they can receive immediate postnatal care and vaccinations for their newborns; (vi) engage with religious leaders and community influencers to promote vaccination as a religious and cultural responsibility; (vii) implement targeted interventions that address the unique socioeconomic and geographic challenges faced by pastoralist communities; (viii) collaborate with local authorities, community leaders, and health extension workers to develop and implement community-based initiatives that are tailored to the specific needs of the Afar region; (ix) allocate sufficient resources to support the implementation of these policies, including budgeting for infrastructure development, personnel, and mHealth initiatives. By addressing these policy implications, the study suggests that it is possible to improve vaccination coverage and reduce the dropout rate among children in the pastoralist communities of Ethiopia, ultimately contributing to better health outcomes and reduced mortality rates.

## 5. Conclusions and Recommendation

The overall proportion of full immunization was 25.1%, which is lower than that of the UNICEF Strategic Plan, 2018–2021, which calls for at least 90% immunization coverage at the national level and at least 80% immunization coverage at the district level. Based on vaccination card observations, the dropout rate from Pentavalet-1 to Pentavalent-3 was found to be 2.9%. Having formal education, owning mobile phones, attending an ANC visit, and birthing in a health facility were positively and significantly associated with full immunization. The study suggests that improving immunization rates can be achieved through several strategies:Promoting Maternal Education—The study found a significant positive association between maternal education and full immunization coverage. Enhancing educational opportunities for mothers and caretakers could lead to a better understanding and prioritization of vaccination.Increasing Mobile Phone Ownership—The study identified a significant link between mobile phone ownership and increased immunization rates. Programs could focus on leveraging mobile technology to disseminate information, send reminders, and provide support related to vaccination.Encouraging ANC visits—The study highlights the importance of ANC visits, showing a strong association with full immunization. Strengthening ANC services and ensuring that more women attend these visits could enhance vaccination coverage.Promoting Health Facility Delivery—The study found that childbirth at a health facility is strongly associated with higher immunization rates. Efforts to increase institutional deliveries could have a positive impact on immunization outcomes.

Overall, the study suggests a multifaceted approach that includes improving access to education, enhancing ANC services, promoting health facility deliveries, and utilizing mobile technology to improve immunization rates in the Afar region. More importantly, The Ethiopian government should focus on improving the level of education among women, as education can contribute to greater vaccination coverage. Additionally, the government and other stakeholders should engage in improving access to and promoting the use of maternal healthcare services such as antenatal clinics and birthing health facilities using different strategies to increase immunization coverage. Generally, HCWs should integrate child vaccination services with other health services, such as maternal health care services. Therefore, the government, NGOs, and other stakeholders should strengthen the expansion of immunization services by increasing and sustaining a focus on outreach programs to hard-to-reach areas of the country. Additionally, our qualitative findings indicate the need for community participation and engagement as well as collaboration between the governmental service delivery system and NGOs in alleviating existing vaccination-related problems such as the construction of health facilities and making services accessible through the mobile health approach. Moreover, the study also indicated that work must be done in relation to motivating health workers, including HEWs.

## Figures and Tables

**Figure 1 ijerph-21-01112-f001:**
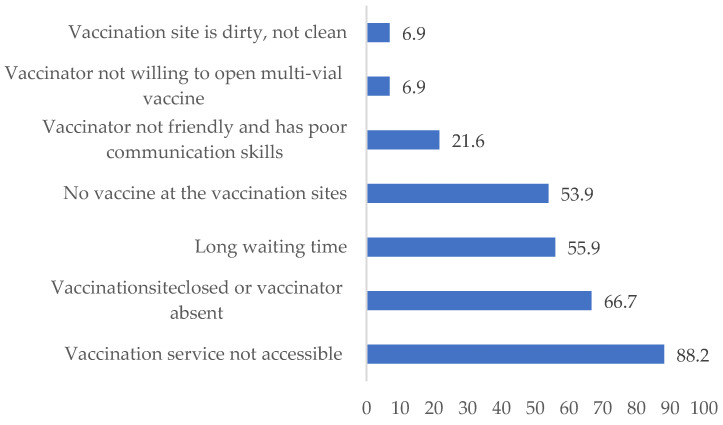
Reasons for dissatisfaction with vaccination services in three woredas of the Afar region.

**Figure 2 ijerph-21-01112-f002:**
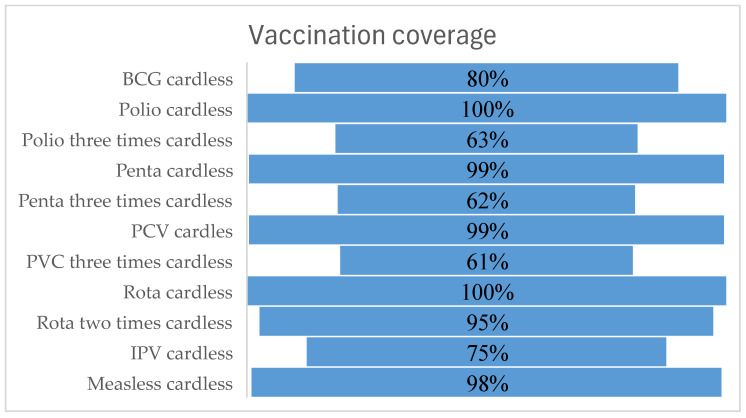
Proportion of basic vaccines based on card observation among children aged 12–23 months in three districts of the Afar region, 2023.

**Table 1 ijerph-21-01112-t001:** Sociodemographic characteristics of respondents in three woredas in the Afar region, 2023.

Variables	Frequency	Percentage (%)
Woreda		
Elidar	154	36.8
Dubti	190	45.3
Gereni	75	17.9
Residence		
Rural	328	78.3
Urban	91	21.7
Age (mean age was 31.8 ± 4)		
<22	80	19.1
22–25	142	33.9
26–29	80	19.1
≥30	117	27.9
Marital status		
Married	408	97.4
Single/divorced/widowed	11	2.6
Maternal educational status		
Unable to read and write	345	82.3
Read and write only	43	10.3
Primary school	17	4.1
Secondary and above	14	3.4
Father’s educational status		
Unable to read and write	296	70.6
Read and write only	58	13.8
Primary school	18	4.3
Secondary school	30	7.2
College and above	17	4.1
Mother’s occupation		
Housewife	331	79.0
Merchant	40	9.5
Farming/pastoralist	29	6.9
Governmental employed	17	4.1
Other specified	2	0.5
Religion status of respondent		
Muslim	398	95
Orthodox	21	5.0
Mass media possession		
Electrical/solar lump (light)	162	38.7
Radio	139	33.2
Television	139	33.2
Mobile phone	307	73.3
Household economic status		
Able to meet basic needs	343	81.9
Unable to meet basic needs without charity	52	12.4
Refuse to answer	14	3.3
Able to meet basic needs and some nonessential goods	9	2.1
Able to purchase most nonessential goods	1	0.2
Total family size (the mean family size of the study was 5.9)		
<4	155	37.0
≥4	264	63.0
Distance to nearest health post		
≤30 min	252	60.1
>30 min	167	39.9
Distance to nearest health center		
≤30 min	137	32.7
>30 min	282	67.3
Distance to the nearest vaccination center		
≤30 min	230	54.9
>30 min	189	45.1

**Table 2 ijerph-21-01112-t002:** Obstetrics and index child characteristics of children studied in three woredas of the Afar region.

Variables	Frequency	Percentage (%)
Gravidity (mean 3.4 ± 1.5)		
≤4	302	72.1
>4	117	37.9
Parity (mean 3.2 ± 1.2)		
≤4	332	79.2
>4	87	20.8
Sex of index child		
Male	235	56.1
Female	184	53.9
Age of indexed child in months (mean 15.2 months ± 5.2)		
12–13	127	30.3
13–16	100	23.9
17–20	89	21.2
21–23	103	24.6
Order of indexed child		
First	48	11.5
Second	81	19.3
Third	115	27.4
Fourth and later	175	41.8
Index childbirth condition		
Single	403	96.2
Twins	16	3.8
Index child living with		
Both parents	399	95.2
Mother only	20	4.8

**Table 3 ijerph-21-01112-t003:** Knowledge and vaccination services related to the characteristics of the study participants in three woredas in the Afar region, 2023.

Variables	Frequency	Percentage (%)
Heard of vaccination		
Yes	313	74.7
No	106	25.3
Advantages of vaccinating children (*n* = 313)		
To protect them from disease	284	90.7
To have healthy child	224	71.6
Have no benefits	2	0.64
Do not know	2	0.64
Disadvantages of vaccine		
Side or adverse effects	210	67.1
May make children sick	178	56.9
Takes time	20	6.4
Other (sterilize children, politics)	12	3.8
Do not know/not sure	50	16
How likely the vaccine is prevented		
Very likely	191	61.0
Somewhat likely	45	14.4
Not likely at all	35	11.2
Do not know/not sure	60	19.2
Seriousness of vaccine preventable diseases		
Very serious	146	46.6
Somewhat serious	91	29.1
Not serious at all	33	10.5
Do not know/not sure	55	17.6
When to start		
At birth	78	24.9
First few weeks	34	10.9
First few months	121	38.7
Later	58	18.5
Do not know	49	15.7
Where to get child vaccine		
Outreach site	139	44.4
Health post	110	35.1
Health center	183	58.5
Public hospital	51	16.3
Received information on vaccination when the child was less than one year (*n* = 313)		
Yes	239	76.4
No	74	23.6
Source of information		
Health professionals (doctors, nurses)	193	80.8
Health extension workers	127	53.1
Radio	20	8.4
Television	48	20.1
Other printed materials (poster, banner)	6	2.5
Husband/partner	113	47.3
Family/friend/neighbor	68	28.5
Religious/community leaders	2	0.8
Type of information heard		
Importance of vaccination	232	97.1
About vaccination campaigns	134	56.1
Where to get routine vaccination	43	18.0
Timing for vaccination	27	11.3
Return to next doses of vaccination	25	10.5
Possible adverse events vaccination	66	27.6
Harms of vaccination	20	8.4
Do not know/not sure	4	1.7
Informed about side effects		
Yes	239	76.4
No	74	23.6
Informed what to do if adverse effects occurred		
Yes	236	98.7
No	3	1.3

**Table 4 ijerph-21-01112-t004:** Factors related to vaccine refusal and vaccination service satisfaction among respondents in the Afar region, 2023.

Characteristics	Frequency	Percentage
Ever refused child vaccine		
Yes	52	21.8
No	261	78.2
Reasons for refusal		
Too many injections at visit	28	53.8
Child was ill	26	50.0
Fear of injection pain	21	40.4
Fear of side effects	11	21.2
Fear of risk of disease transmission	6	11.5
Doubts on the benefit of the vaccine	6	11.5
Has already been vaccinated	4	7.7
Fear, doubts, suspicions about vaccine (*n* = 313)		
Yes	27	8.6
No	278	88.8
Not sure	8	2.6
Reasons for fear, doubts, suspicions about vaccine (*n* = 27)		
Vaccinations cause side effects	15	55.6
Vaccinations can make children sick	8	29.6
Vaccinations sterilize children	2	7.4
Others (politics, religious)	2	7.4
Vaccine recommendation to other community members		
Yes	271	86.6
No	42	13.4
Reasons not to recommend to others		
Do not believe vaccines are useful	30	71.4
Causes side effects/makes them sick	29	69
Injection can transmit diseases	9	21.4
Against social/religious norm	1	2.4
Satisfaction of vaccination service		
Yes	211	67.4
No	102	32.6

**Table 5 ijerph-21-01112-t005:** Maternal health service utilization in three woredas in the Afar region, 2023.

ANC Follow-Up	Frequency	Percentage
Yes	240	57.3
No	179	42.7
Place of ANC booking		
Health post	66	27.5
Health center	152	63.3
Public hospital	67	27.9
ANC frequency		
≤3	237	98.8
>3	3	1.2
Information about child vaccination during ANC		
Yes	207	86.3
No	25	10.4
Not sure/do not know	8	3.3
Place of birth		
Home	227	54.2
Health facility	192	45.8
Birth assistance		
Health professionals including HEWs	192	45.8
Traditional birth attendant	214	51.1
Family/friend/neighbor	11	2.6
No one	2	0.5
Check-up after birth		
Yes	103	24.6
No	316	75.4
Who made the check-up		
Health professionals	93	90.3
Health extension workers	9	8.7
Traditional birth attendant	1	0.9
Information on vaccination during check-up		
Yes	97	94.2
No	4	3.9
Not sure/do not know	2	1.9

**Table 6 ijerph-21-01112-t006:** Proportion of basic vaccines based on card observation among children aged 12–23 months in three districts of the Afar region, 2023.

Vaccines	Proportion (Total *n* = 92)	95% CI
BCG	97.8% (*n* = 90)	91.6–99.5%
Polio-1	98.9% (*n* = 91)	92.5–99.9%
Penta-1	96.7% (*n* = 89)	90.2–98.9%
PCV-1	97.8% (*n* = 90)	91.6–99.5%
IPV at 14 weeks	88.1% (*n* = 81)	79.5–93.3%
Polio-2	90.2% (*n* = 83)	82.1–94.9%
Penta-2	90.2% (*n* = 83)	82.1–94.9%
PCV-2	89.1% (*n* = 82)	80.8–94.1%
Polio-3	84.8% (*n* = 78)	80.8–94.1%
Penta-3	83.7% (*n* = 77)	79.5–93.3%
PCV-3	82.6% (*n* = 76)	80.8–94.1%
Rota-1	97.8% (*n* = 90)	91.6–99.5%
Rota-2	91.3% (*n* = 84)	83.4–95.6%
Measles at 9 months	91.3% (*n* = 84)	83.4–95.6%

**Table 7 ijerph-21-01112-t007:** Barriers and reasons for not vaccinating children among study participants in the Afar region.

Variables	Frequency	Percentage (%)
Why children do not vaccinate		
Health workers did not come to the village	247	58.9
Domestic workload	180	43.0
Vaccination service not accessible	155	37.0
Vaccination site closed/vaccinator absent	153	36.5
Long waiting time	138	32.9
No vaccine at the vaccination sites	106	25.3
My husband discouraged me	71	16.9
Vaccination makes them sick	33	7.9
Vaccinator not friendly/poor relationship	32	7.6
Family/community discouraged me	27	6.4
Cultural or religious norms or beliefs	14	3.3
Vaccine approval status		
Yes	308	73.5
No	85	20.3
Not sure/do not know	26	6.2
Approved positively by		
Husband/partner	301	97.7
Parents/parents-in-laws	134	43.5
Neighbors/peers	105	34.1
Other family members	27	8.8
Difficulty in remembering the vaccination schedule (*n* = 247)		
Not difficult at all	51	20.7
Somewhat difficult	52	21.1
Very difficult	161	65.2
Cultural taboos against vaccinating (*n* = 419)		
Yes	66	15.8
No	328	78.3
Do not know/not sure	25	6.0

**Table 8 ijerph-21-01112-t008:** Bivariable and multivariable logistic regression analysis for full vaccination among children aged 12–23 months in the Afar region, 2023.

Variables	Fully Immunized		Crude Odds Ratio	Adjusted Odds Ratio
	Yes	No		
Health center distance				
<30 min	55	82	3.11 (197–4.92)	1.76 (0.40–1.41)
≥30 min	50	232	1	1
Owning a mobile phone				
Yes	95	212	4.57 (2.29–9.14)	2.99 (1.33–6.76) **
No	10	102	1	1
Maternal education				
Formal education	23	2	10.73 (4.63–24.87)	3.90 (1.53–9.98) **
No formal education	82	306	1	1
Child’s age				
12–15	42	156	0.62 (0.37–1.04)	0.66 (0.35–1.25)
16–19	26	73	0.82 (0.45–1.48)	0.61 (0.31–1.23)
20–23	37	85	1	
ANC visit				
Yes	91	149	7.19 (3.93–13.18)	2.39 (1.14–5.01) **
No	14	165	1	1
Place of birth				
Health facility	87	105	9.62 (5.50–16.83)	5.79 (2.77–12.12) **
Home	18	209	1	1
Birth order				
First	15	33	1.64 (0.81–3.33)	0.99 (0.42–2.36)
Second	23	58	1.43 (0.78–2.61)	1.32 (0.62–2.79)
Third	29	86	1.22 (0.69–2.11)	1.08 (0.57–2.07)
Fourth and later	38	137	1	
Information received during post-natal period				
Yes	38	65	2.17 (1.34–3.52)	0.74 (0.39–1.38)
No	67	249	1	

** represents *p* < 0.01.

## Data Availability

The data presented in this study are available based on request from the corresponding authors. The data are not publicly available due to privacy reasons.

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
