# Peer review of "Vaccination Coverage and Predictors of Vaccination among Children Aged 12–23 Months in the Pastoralist Communities of Ethiopia: A Mixed Methods Design"

_ijerph, 2024, doi:10.3390/ijerph21081112_

Round 1

Reviewer 1 Report

Comments and Suggestions for Authors

I recently read your manuscript, "Vaccination Coverage and Predictors of Vaccination Among Children Aged 12-23 Months in the Pastoralist Communities of Ethiopia: A Mixed Methods Design," with great interest. This study provides valuable insights into vaccination coverage, dropout rates, and associated factors among children in the Afar Region of Ethiopia. It offers a significant contribution to understanding the dynamics of immunization in pastoralist communities.

To further enhance the clarity and impact of your manuscript, I would like to suggest a few minor revisions and pose several questions for your consideration:

  1. Could you provide a more detailed explanation of what makes this study innovative or how it contributes new insights to the field?
  2. In the abstract and introduction, please concisely state the specific objectives of your study.
  3. What was the vaccination coverage percentage among the children surveyed?
  4. What method was used to select participants for the study?
  5. Which factors were significantly associated with full vaccination rates?
  6. What was the dropout rate from Pentavalent-1 to Pentavalent-3?
  7. How does this study suggest improving immunization rates?
  8. How does maternal education impact vaccination rates?
  9. What role do antenatal care (ANC) services play in influencing vaccination rates?
  10. How can mHealth interventions improve vaccination coverage?
  11. I suggest insert/use/discuss some more recent data (Short sentences)

  - DOI: 10.3390/vaccines11091458

  1. Could you provide more data or discussion on the importance of healthcare professionals worldwide recognizing the need for targeted interventions that address educational gaps, improve access to health services, and utilize technology to enhance vaccination coverage among children in Ethiopia's pastoralist communities?

These revisions and additions will strengthen the manuscript and provide a clearer understanding of the research's implications. I look forward to reviewing the updated version of your work.

Comments on the Quality of English Language

Minor editing of English language required

Author Response

We greatly appreciate your invaluable feedback. We have revised the main manuscript according to your feedback and the feedback from the other reviewers. We would be grateful if you could review the changes we have made to the paper. Additionally, please find below a point-by-point response to the feedback we received.

  1. Could you provide a more detailed explanation of what makes this study innovative or how it contributes new insights to the field?

Thank you for the feedback.   Some of the key things that stand out about how this study contributes new insights include the focused examination of vaccination in a pastoralist population using mixed methods, the identification of specific predictors, and the geographic concentration on the understudied Afar region of Ethiopia. This contributes important new knowledge to improve vaccination access and uptake in hard-to-reach communities.

2. In the abstract and introduction, please concisely state the specific objectives of your study.

Thank you for the feedback. In the abstract, we mentioned the objective as follow” This study assessed vaccination coverage and associated factors among children aged 12-23 months in the pastoralist Ethiopia.”

3. What was the vaccination coverage percentage among the children surveyed?

Thank you for the feedback. The percentage of children who received full vaccination was 25%. Based on vaccination card observations, the dropout rate from Pentavalent-1 to Pentavalent-3 was found to be 2.9%.

4. What method was used to select participants for the study?

Thank you for the feedback. In the method section under the sample size and sampling procedure See “This sample was withdrawn from the three identified districts; in each district, three to four kebeles, the smallest administrative units in Ethiopia, were randomly selected using the WHO-recommended survey strategy approach for immunization. Once the kebeles were identified, the Gote/Villages were mapped. In the village, all candidate households (12-23 months and mother/father/caretaker pairs) were listed using data from HEWs data registry document, and simple random sampling approaches were used to select study participants.”

5. Which factors were significantly associated with full vaccination rates

Thank you for the feedback. The following were significantly associated with full vaccination rates: Having formal education 3.90 (1.53-9.98), owning mobile phones 2.99 (1.33-6.76), attending an ANC visit 2.39 (1.14-5.01) and birthing in a health facility 5.79 (2.77-12.12) were positively and significantly associated with full immunization.

6. What was the dropout rate from Pentavalent-1 to Pentavalent-3?

Thank you for the feedback. Based on vaccination card observations, the dropout rate from Pentavalent-1 to Pentavalent-3 was found to be 2.9%.

7. How does this study suggest improving immunization rates?

Thank you for the feedback. The study suggests that improving immunization rates can be achieved through several strategies:

  • Promoting Maternal Education- The study found a significant positive association between maternal education and full immunization coverage. Enhancing educational opportunities for mothers and caretakers could lead to better understanding and prioritization of vaccination.
  • Increasing Mobile Phone Ownership-The study identified a significant link between mobile phone ownership and increased immunization rates. Programs could focus on leveraging mobile technology to disseminate information, send reminders, and provide support related to vaccination.
  • Encouraging Antenatal Care (ANC) Visits- The study highlights the importance of ANC visits, showing a strong association with full immunization. Strengthening ANC services and ensuring that more women attend these visits could enhance vaccination coverage.
  • Promoting Health Facility Delivery- The study found that childbirth at a health facility is strongly associated with higher immunization rates. Efforts to increase institutional deliveries could have a positive impact on immunization outcomes.

Overall, the study suggests a multifaceted approach that includes improving access to education, enhancing ANC services, promoting health facility deliveries, and utilizing mobile technology to improve immunization rates in the Afar Region.

8. How does maternal education impact vaccination rates?

Thank you for the feedback. The findings suggest that complete immunization was more common in children whose mothers were educated, partly because maternal education leads to the acquisition of literacy skills that can be applied to knowledge of vaccination and child protection and better health-seeking behaviours, which then improves immunization uptake for their children. This suggests that educated mothers are better informed about the importance of vaccinations and the negative impact of unvaccinated baby, the diseases they prevent, and the schedules for immunization, more likely to access healthcare services, and more proactive in seeking out and adhering to immunization schedules

9. What role do antenatal care (ANC) services play in influencing vaccination rates?

Thank you for the feedback. During ANC visits, healthcare providers often educate expectant mothers about the importance of child immunization, the schedule of vaccines, and the diseases they prevent, linking mothers to the healthcare system, and facilitating follow-up for post-delivery vaccinations. These visits often integrate other maternal and child health services, fostering continued access to healthcare, which increases trust and confidence in the system. Additionally, ANC services promote institutional deliveries, which are associated with higher immunization rates, thereby indirectly improving overall vaccination coverage.

10. How can mHealth interventions improve vaccination coverage?

Thank you for the feedback. We revised accordingly and add in the revised section

In Ethiopia, where challenges such as geographic barriers, limited healthcare infrastructure, and low literacy levels often hinder vaccination efforts, mHealth initiatives can play a crucial role. By providing timely SMS reminders to parents about upcoming vaccination appointments, mHealth can help reduce missed opportunities and dropouts, especially in remote areas. These interventions can also deliver culturally relevant educational content, increasing awareness and understanding of vaccines, which is essential in combating misinformation. Furthermore, mHealth can facilitate real-time data collection and tracking, allowing healthcare providers to monitor vaccination coverage and quickly identify regions with low uptake. By improving access to information and services, mHealth can help overcome the unique barriers faced in Ethiopia, leading to higher vaccination rates and better overall health outcomes.

11. I suggest insert/use/discuss some more recent data (Short sentences)

  - DOI: 10.3390/vaccines11091458

Thank you for the feedback. We have included this.

12. Could you provide more data or discussion on the importance of healthcare professionals worldwide recognizing the need for targeted interventions that address educational gaps, improve access to health services, and utilize technology to enhance vaccination coverage among children in Ethiopia's pastoralist communities?

Thank you for your thoughtful question. We agree that healthcare professionals globally must recognize the critical need for targeted interventions in Ethiopia's pastoralist communities to enhance vaccination coverage. we enhance our discussion as follow to better illustrate the global relevance of these targeted interventions and the crucial role healthcare professionals play in implementing them to improve vaccination outcomes in Ethiopia's pastoralist communities. kindly find in the discussion section line 463- 484.

The unique challenges faced by pastoralist communities in Ethiopia, such as geographic isolation, nomadic lifestyles, and limited access to healthcare infrastructure, create substantial barriers to routine immunization. These challenges are well-documented in studies of similar populations, where the combination of mobility, remote locations, and the scarcity of healthcare facilities complicates efforts to deliver consistent health services. https://www.researchgate.net/publication/10654721_Assessment_of_immunization_service_in_perspective_of_both_the_recipients_and_the_providers_a_reflection_from_focus_group_discussions

These barriers underscore the necessity of targeted interventions that are specifically designed to address the distinctive needs of these communities.

One critical area for intervention is the educational gap that often exists in these regions. Culturally tailored health education programs that engage local leaders and utilize local languages are essential for effectively communicating the importance of vaccination. Such programs have been shown to improve health outcomes by increasing awareness and changing attitudes toward immunization within similar contexts https://www.ncbi.nlm.nih.gov/pmc/articles/PMC2672574/

By involving respected community figures and delivering messages in a culturally relevant manner, these programs can significantly enhance the uptake of vaccinations.

In addition to educational initiatives, improving access to healthcare services through innovative delivery methods is crucial. Mobile clinics and outreach programs have been successfully implemented in other low-resource settings to bring essential services directly to hard-to-reach populations. Moreover, mHealth initiatives—such as mobile phone reminders and health information dissemination—offer a promising way to maintain contact with nomadic populations and ensure timely vaccination https://pubmed.ncbi.nlm.nih.gov/25881735/

By leveraging technology and flexible service delivery models, healthcare providers can overcome many of the logistical challenges associated with serving pastoralist communities

Reviewer 2 Report

Comments and Suggestions for Authors

This paper investigates the factors contributing to mothers' decisions regarding their children's vaccinations during the infant period. The paper utilized a mixed-method approach to explore the association and context of these decisions. For the quantitative part, the authors used a survey designed by WHO, while the qualitative part was conducted through semi-structured interviews. The results indicate a potential improvement in vaccination coverage compared to previous literature. Furthermore, it shows that mothers' literacy and place of birth are key variables associated with vaccination decisions.

This paper is simple yet informative. The authors properly deployed statistical tools to support their results and discussion. However, the novelty of the paper is a concern.

Major Revision:

·         Lack of novelty: The results from this study are generally reflected in many other studies. Conceptually, this paper discusses mothers’ socioeconomic status and their children's vaccination.

Examples found include:

o    van den Boogaard, J., Rots, N. Y., van der Klis, F., de Melker, H. E., & Knol, M. J. (2020). Is there an association between socioeconomic status and immune response to infant and childhood vaccination in the Netherlands?. Vaccine, 38(18), 3480-3488.

o    Anello, P., Cestari, L., Baldovin, T., Simonato, L., Frasca, G., Caranci, N., ... & Canova, C. (2017). Socioeconomic factors influencing childhood vaccination in two northern Italian regions. Vaccine, 35(36), 4673-4680.

o    Hajizadeh, M. (2018). Socioeconomic inequalities in child vaccination in low/middle-income countries: what accounts for the differences?. J Epidemiol Community Health, 72(8), 719-725.

o    Nankabirwa, V., Tylleskär, T., Tumwine, J. K., Sommerfelt, H., & Promise-ebf Study Group Thorkild. Tylleskar@ cih. uib. no. (2010). Maternal education is associated with the vaccination status of infants less than 6 months in Eastern Uganda: a cohort study. BMC Pediatrics, 10, 1-9.

·         Therefore, the authors must reconsider the novelty of the paper. One potentially interesting point could be the differences in sample group locations and how these factors contribute to the findings.

Minor Revision:

  • Figure 2: The authors should remake the figure to be more readable. Proper labels are needed. For example, "BCG_Cardless" should be "BCG Cardless," and the y-axis should be labeled "Percent of samples." Additionally, the authors should provide more context about the term "cardless."
  • Table 8: An explanation about the p-value is missing. The authors used asterisks (*) to show significance levels but did not explain the range of significance. For instance, *** represents p<0.001, ** represents p<0.01, etc.

Author Response

Reviewer 2

We greatly appreciate your invaluable feedback. We have revised the main manuscript according to your feedback and the feedback from the other reviewers. We would be grateful if you could review the changes we have made to the paper. Additionally, please find below a point-by-point response to the feedback we received.

Comments and Suggestions for Authors

This paper investigates the factors contributing to mothers' decisions regarding their children's vaccinations during the infant period. The paper utilized a mixed-method approach to explore the association and context of these decisions. For the quantitative part, the authors used a survey designed by WHO, while the qualitative part was conducted through semi-structured interviews. The results indicate a potential improvement in vaccination coverage compared to previous literature. Furthermore, it shows that mothers' literacy and place of birth are key variables associated with vaccination decisions.

This paper is simple yet informative. The authors properly deployed statistical tools to support their results and discussion. However, the novelty of the paper is a concern.

Major Revision:

  • Lack of novelty: The results from this study are generally reflected in many other studies. Conceptually, this paper discusses mothers’ socioeconomic status and their children's vaccination.

Examples found include:

o    Van den Boogaard, J., Rots, N. Y., van der Klis, F., de Melker, H. E., & Knol, M. J. (2020). Is there an association between socioeconomic status and immune response to infant and childhood vaccination in the Netherlands?. Vaccine, 38(18), 3480-3488.

o    Anello, P., Cestari, L., Baldovin, T., Simonato, L., Frasca, G., Caranci, N., ... & Canova, C. (2017). Socioeconomic factors influencing childhood vaccination in two northern Italian regions. Vaccine, 35(36), 4673-4680.

o    Hajizadeh, M. (2018). Socioeconomic inequalities in child vaccination in low/middle-income countries: what accounts for the differences? J Epidemiol Community Health, 72(8), 719-725.

o    Nankabirwa, V., Tylleskär, T., Tumwine, J. K., Sommerfelt, H., & Promise-ebf Study Group Thorkild. Tylleskar@ cih. uib. no. (2010). Maternal education is associated with the vaccination status of infants less than 6 months in Eastern Uganda: a cohort study. BMC Pediatrics, 10, 1-9.

Therefore, the authors must reconsider the novelty of the paper. One potentially interesting point could be the differences in sample group locations and how these factors contribute to the findings.

Thank you for your valuable feedback. We appreciate the observations regarding the study's novelty. To address the concern, we will emphasize the unique aspects of our research and differentiate it from existing studies. Our study provides new insights into how specific regional factors within Ethiopia, particularly in the Afar region's pastoralist communities, influence vaccination coverage. Unlike the studies cited, which focus on relatively more stable settings with established healthcare systems, our research highlights the challenges faced in a region characterized by nomadic lifestyles and significant geographic and socioeconomic barriers.

Moreover, in the pastoral context, such as in the Afar regions, there is a scarcity of evidence globally, and particularly in the Ethiopian context. The findings from these regions are highly specific and only applicable in this context. For instance, mHealth (mobile health) is an appropriate approach for these mobile communities where it is challenging to provide services in a static context, this might be different for where context is different. Furthermore, we elaborate on how the unique socio-cultural and logistical conditions of the Afar region contribute to the findings, providing a comparative analysis with other regions and settings. This approach underscores the specific context of our study and its contributions to the broader understanding of vaccination coverage in underserved, nomadic populations. In line with this, we included a detailed discussion on the implications of these regional differences for policy and practice, making the case for tailored interventions that address the unique needs of such communities. This should enhance the paper’s originality and relevance.

Minor Revision:

Figure 2: The authors should remake the figure to be more readable. Proper labels are needed. For example, "BCG_Cardless" should be "BCG Cardless," and the y-axis should be labeled "Percent of samples." Additionally, the authors should provide more context about the term "cardless."

Thank you for your feedback. We revised the figures and the labeling

  •  

Table 8: An explanation about the p-value is missing. The authors used asterisks (*) to show significance levels but did not explain the range of significance. For instance, *** represents p<0.001, ** represents p<0.01, etc.

Thank you for your suggestion. Clearly show p value ranges in the full documents which is actually ** p<0.01

Reviewer 3 Report

Comments and Suggestions for Authors

1) At the end of the introduction section, is not clear the aim or primary objetive of de study. The information it is there (lines 84-97), but is not clear what they want to analyze and why. 2) The results from Afar could be extrapolable to all the country? Why yes or why not? Please make a detailed explanation about that point. This questions are related with results from section 3.1 3) Which ethical committee have approve the protocol? Please provide this information. This is a very important point that need to be added to the manuscript. 4) In the introduction section is not well describe the roll of religion and vaccination, please provide some context about that. 5) About the number of family members, the authors reported as a categorical variable, more or less than 4, please provide information with the mean or median (according data distribution). 6) The translate to english in some sections is not enough correct, for example line 201. Please improve english grammar. 7) Same comment about parity, gravity and age. Please provide information as a numerical variable (mean or median, according data distribution). 8) There are a similar study to compare results from table 3? 9) Please provide a deeper explanation about the results of Figure 1. Why vaccination was no available or vaccinator was absent? 10) Please compre table 6 (in discussion) with other coverage of other similar countries. 11) Which is the differences between results on table 6 and the data reported to WHO.? 12) Improve figure 2, you can not have more than 1. Also chance absolute numbers to % 13) Results of table 7 can be related or explained by the religion? 14) Extend your discussion section.

Comments on the Quality of English Language

The translate to english in some sections is not enough correct, for example line 201. Please improve english grammar. 

Author Response

to your feedback and the feedback from the other reviewers. We would be grateful if you could review the changes we have made to the paper. Additionally, please find below a point-by-point response to the feedback we received.

Comments and Suggestions for Authors

  • At the end of the introduction section, is not clear the aim or primary objective of de study. The information it is there (lines 84-97) but is not clear what they want to analyze and why.

Thank you for the feedback. We acknowledge the need to clarify the primary objective of our study at the end of the introduction section. To address this, we will revise the relevant portion to explicitly state the study's aim and its significance.

We have incorporated the following ‘’ The primary objective of this study is to assess vaccination coverage, the dropout rate, and the factors associated with full immunization among children aged 12-23 months residing in the Afar Region, Ethiopia. This research is significant because it addresses a clear gap in the literature, focusing on the unique sociocultural, economic, and geographical factors that influence vaccination behavior in this under-researched area. Some of the key things that stand out about how this study contributes new insights include the focused examination of vaccination in a pastoralist population using mixed methods, the identification of specific predictors, and the geographic concentration on the understudied Afar region of Ethiopia. This contributes important new knowledge to improve vaccination access and uptake in hard-to-reach communities. By employing a mixed methods design that integrates both quantitative and qualitative approaches, our study aims to provide a comprehensive and nuanced understanding of the determinants of vaccination in the Afar region, ultimately contributing valuable insights that can inform targeted interventions to improve immunization coverage in similar contexts.”

  • The results from Afar could be extrapolable to all the country? Why yes or why not? Please make a detailed explanation about that point. These questions are related with results from section 3.1

Thank you for the feedback. The results from our study in the Afar Region are not directly extrapolable to the entire country of Ethiopia due to significant regional differences in sociocultural, economic, and geographic contexts. Ethiopia is a highly diverse country with various regions characterized by distinct lifestyles. Afar, as a predominantly pastoralist and semi-nomadic region, presents unique challenges that may not be present in other parts of Ethiopia, such as the high mobility of its population, limited healthcare facilities, and unique cultural practices. These factors heavily influence vaccination behaviors and coverage in ways that differ from other regions, such as the urban centers or agrarian communities in other parts of the country. The specific findings from Afar may not be fully generalizable to all of Ethiopia, they provide critical insights into the challenges faced by other pastoralist and remote communities within the country. These insights are valuable for understanding and addressing vaccination coverage in similar contexts both within Ethiopia and in other countries with pastoralist populations.

  • Which ethical committee have approve the protocol? Please provide this information. This is a very important point that need to be added to the manuscript.

Thank you for the feedback.  The research secured an ethical clearance granted from Amref Health Research council in collaboration Afar regional health bureau with the support of Semerha University (Ref No ETCO/Admin/267/23/).

  • In the introduction section is not well describe the role of religion and vaccination, please provide some context about that.

Thank you for the feedback. We incorporate the influence of religion on vaccination as the following. Kindly please find the revised part in the introduction section, line 80- 97.

Religion can significantly influence health behaviours, including decisions related to vaccination. In many communities, religious beliefs and practices shape attitudes toward health interventions, often determining whether individuals and families choose to vaccinate their children. Some religious groups view vaccination positively, seeing it as a means to protect health, which aligns with their beliefs about caring for the body as a sacred responsibility https://pubmed.ncbi.nlm.nih.gov/23499565/

Conversely, other groups may express hesitancy or resistance toward vaccination based on religious or cultural reasons, which can create barriers to immunization https://pubmed.ncbi.nlm.nih.gov/21664679/

In Ethiopia, a country with a diverse religious landscape, the impact of religion on vaccination practices is particularly pronounced. Religious leaders often hold significant influence over their communities and can sway public opinion on health matters. In some areas, including the Afar region, religious and cultural factors may contribute to lower vaccination rates. The predominantly Muslim population in Afar may have unique religious practices and beliefs that affect their healthcare-seeking behaviors, including the acceptance or refusal of vaccines https://www.researchgate.net/publication/374117793_Immunization_of_Children_in_Africa_Strides_and_Challenges

Engaging religious leaders in public health campaigns and incorporating religious teachings that support vaccination can help increase vaccine acceptance in communities where religious beliefs are a significant factor. This approach is especially important in regions like Afar, where traditional and religious practices are deeply rooted in daily life.

  • About the number of family members, the authors reported as a categorical variable, more or less than 4, please provide information with the mean or median (according to data distribution).

Thank you for the feedback.  The mean family size of the study was 5.9

  • The translate to english in some sections is not enough correct, for example line 201. Please improve english grammar.

Thank you for the feedback. We have corrected the English grammar as the following.

Less than three-quarters of the mothers had a gravidity of four or fewer (302, 72.1%).

  • Same comment about parity, gravity and age. Please provide information as a numerical variable (mean or median, according to data distribution).

Thank you for the feedback. We have revised all those variables and provided the mean and standard deviation for all continuous variables in the revised manuscript.

  • Please provide a deeper explanation about the results of Figure 1. Why vaccination was no available or vaccinator was absent?

Thank you for the feedback. The absence of vaccinators in the Afar region highlights significant human resource challenges, including shortages of trained healthcare workers, high turnover, and absenteeism due to harsh living conditions and limited career prospects. Additionally, operational difficulties such as inadequate transportation and security concerns further hinder the presence of vaccinators, exacerbating the region's struggle to maintain consistent immunization services.

  • Please compare table 6 (in discussion) with other coverage of other similar countries. Which is the difference between results on table 6 and the data reported to WHO.?

Thank you for the feedback. We revised the discussion with this and the other reviewers

  • Improve figure 2, you cannot have more than 1. Also chance absolute numbers to %

Thank you for the feedback.  We revised and included in the revised manuscript

  • Results of table 7 can be related or explained by the religion?

Thank you for the feedback. Religious beliefs and practices can sometimes act as barriers to vaccinating children, either directly influencing parents' decisions or indirectly through community norms and pressures.

  • Extend your discussion section. Thank you for the feedback. We have revised the discussion based on the three reviewers. We add and expand the discussion very well.

Round 2

Reviewer 3 Report

Comments and Suggestions for Authors

I am glad with all the modifications.